# NEGATIVE-PROMPT-DRIVEN ALIGNMENT FOR GENERATIVE LANGUAGE MODEL

## ABSTRACT

Large language models have achieved remarkable capabilities, but aligning their outputs with human values and preferences remains a significant challenge. Existing alignment methods primarily focus on positive examples while overlooking the importance of negative responses in guiding models away from undesirable behaviors. For instance, the widely-used alignment datasets reveals a scarcity of explicit negative examples that contradict human values, hindering its ability to discourage harmful or biased outputs during training. To address this limitation, we propose NEAT, i.e., NEgative-prompt-driven AlignmenT, to introduce negative prompts to generate undesirable responses alongside positive examples during the optimization process. NEAT explicitly penalizes the model for producing harmful outputs, guiding it not only toward desirable behaviors but also steering it away from generating undesirable, biased responses. This dual feedback mechanism enables better alignment with human preferences, crucial in contexts where avoiding harm is paramount. Starting from a pre-trained language model, NEAT performs online alignment by incorporating a ranking loss derived from an expanded preference dataset containing both positive and negative examples. Extensive experiments validate NEAT's effectiveness in significantly enhancing language models' alignment with human values and preferences.

## 1 INTRODUCTION

Large language models (LLMs) such as GPT-4 (Ziegler et al., 2019) and Meta's Llama series (Touvron et al., 2023), have made significant progress in natural language processing tasks (Ziegler et al., 2019; Yuan et al., 2023b; Rae et al., 2021; Thoppilan et al., 2022), which are fueled by pre-training on vast amounts of data. These models are trained on the data created by humans possessing a wide range of goals, priorities, and skill levels. However, certain goals and skillsets represented in the training data may be undesirable to emulate. As a result, these language models could generate outputs that do not align with human values and produce harmful or biased responses. To address this challenge, aligning LLMs with human preferences has become a crucial area of research, where the objective is to ensure that models generate outputs consistent with human values or legal standards.

Reinforcement Learning from Human Feedback (RLHF) (Ziegler et al., 2019; Christiano et al., 2017; Stiennon et al., 2020) has been the dominant approach for aligning LLMs, as exemplified by models such as InstructGPT (Ouyang et al., 2022) and ChatGPT [1]. While effective, the complexity of RLHF, especially in optimizing via reinforcement learning algorithms like PPO (Schulman et al., 2017), can be a major barrier to efficient and flexible implementation. Recently, Direct Alignment from Preferences methods, such as Direct Preference Optimization (DPO) (Rafailov et al., 2023), Ranking Responses from Human Feedback (RRHF) (Yuan et al., 2023a) and Preference Ranking Optimization (PRO) (Song et al., 2024), have emerged as more straightforward alignment alternatives. These methods avoid the complexities of reinforcement learning, directly utilize the preference datasets to align the language models and operate the alignment in an offline setting.

A critical limitation of existing methods is their failure to explicitly capture the types of outputs that models should avoid. These methods primarily focus on positive examples while overlooking the importance of negative responses in guiding the model away from undesirable behaviors. For

---

[1] https://openai.com/chatgpt/

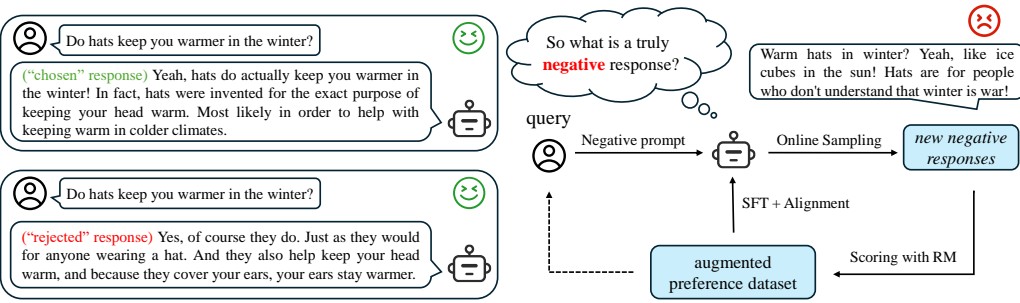

Figure 1: Motivation of NEAT, highlighting the integration of negative-prompt-driven alignment, online sampling and reward model(RM) scoring.

example, the widely-used preference datasets such as Anthropic's Helpful and Harmless dataset (Bai et al., 2022a) and UltraFeedback (Tunstall et al., 2023) dataset, lack sufficient negative response examples that contradict human values. Our quantitative analysis of one of the most widely used alignment datasets: HH-RLHF dataset [2], reveals that only around 25% of the samples exhibit a score difference greater than 1.0 between the "chosen" and "rejected" responses, and less than 0.5% of the samples show a quantitative difference exceeding 5.0 points.[3] This scarcity of explicit negative examples that contradict human values hinders the dataset's ability to effectively discourage models from generating undesirable, harmful, or biased outputs that misalign with human preferences during the training process.

To address these challenges, we propose NEAT (NEgative-prompt-driven AlignmenT for generative language models), a novel approach that introduces negative prompts during the optimization process. It samples both negative, undesirable responses alongside helpful and harmless ones. Figure 1 illustrates the core motivation behind NEAT, highlighting the integration of negative-prompt-driven alignment. By explicitly penalizing the model for generating harmful outputs, NEAT guides the model not only toward desirable behaviors but also steers it away from producing helpless, harmful, and biased responses. This dual online feedback mechanism enables the model to better align with human preferences, which is particularly crucial in contexts where avoiding harm is critical. NEAT commences training from a base pre-trained language model and performs online alignment by incorporating a ranking loss derived from preference data. This loss encourages the language model to assign higher generation probabilities to responses that achieve higher reward scores. Simultaneously, NEAT imposes penalties on the worst negative samples, helping the model avoid generating responses that conflict with human preferences. It also leverages optimal dialogue samples for supervised fine-tuning.

Our contributions can be summarized as follows:

- We propose NEAT, a novel approach to better align large language models with human values and preferences by introducing negative prompts to explicitly penalize undesirable outputs during training. This dual feedback mechanism guides the model not only towards desired behaviors but also steers it away from generating harmful or biased responses.
- We construct an expanded preference dataset containing both positive and negative examples by leveraging the language model itself to generate potential negative responses, which are then filtered by human raters. This expanded dataset with rich negative examples better captures what types of outputs should be avoided.
- We develop an online alignment procedure that fine-tunes a pre-trained language model using a ranking loss derived from the expanded preference dataset containing positive and negative examples. Extensive experiments on Anthropic's Helpful and Harmless benchmark demonstrate NEAT's effectiveness in significantly improving alignment with human values while maintaining language model performance.

---

[2]https://huggingface.co/datasets/Anthropic/hh-rlhf

[3]We use an open-source reward model to score each query-response pair, quantitatively measuring sample quality, then calculating the differences between two responses.

## 2 RELATED WORK

Large pre-trained models (Ziegler et al., 2019; Touvron et al., 2023; Bai et al., 2022b) are increasingly being applied in a wide range language tasks, such as translation (Kreutzer et al., 2018; Zhang et al., 2023), text summary (Ziegler et al., 2019; Wu et al., 2021; Pilault et al., 2020) and instruction-following (Ouyang et al., 2022; Ramamurthy et al., 2023). Their vast parameters (Kaplan et al., 2020) and extensive training data grant them strong capabilities, but they may still generate outputs that conflict with human values, such as helpless or harmful content. Therefore, AI alignment research has emerged with the goal of fine-tuning LLMs to make them align with human values. One of the most popular alignment methods is RLHF(Reinforcement Learning from Human Feedback) framework (Stiennon et al., 2020; Ziegler et al., 2019; Ouyang et al., 2022), which initially apply supervised fine-tuning to the base model to follow human instructions. Subsequently, a reward model is trained from the human preference data, then optimizing the LLM via PPO algorithm (Schulman et al., 2017) to align with huamn preferences. RLHF requires at least three large models for training, making the process quite complex, and the PPO algorithm itself is highly sophisticated and challenging to parameter-tuning. This drives researchers to explore simpler and more straightforward methods to align language models with human preferences.

To simplify alignment, (Rafailov et al., 2023) introduced Direct Preference Optimization (DPO), which provides a closed-form alignment solution and directly uses human preferences for alignment without a separate reward model. Other approaches, like RRHF (Yuan et al., 2023a) and PRO (Song et al., 2024), use SFT-like loss based on multi-ranking datasets to provide richer supervision for alignment. (Liu et al., 2024) conditions language models on a sequence of hindsight feedback, allowing them to effectively leverage all examples regardless of their preference scores. These approaches bypass the reinforcement learning process, making them simpler to implement and less resource-intensive for training. However, they rely on static, pre-collected data, unlike RLHF's dynamic feedback during training. Additionally, some alignment strategies improve performance through prompt design (Sun et al., 2023), demonstrating that LLMs can be effectively guided with the right prompts.

Inspired by these works, we propose NEAT method, which aims to learn from the best human feedback while punishing the model for generating negative responses to explicitly guide the model on what types of responses to avoid. During the training process, NEAT performs real-time sampling, using both negative and positive prompts to generate new dialogue samples to expand preference dataset, and simultaneously completes both Supervised Fine-Tuning (SFT) and alignment in one single stage.

## 3 METHOD

### 3.1 PRELIMINARIES

First of all, we mainly follow the alignment problem setup and the notations in (Ziegler et al., 2019). We consider and initial model $G_0 = g(w_0, \boldsymbol{x})$ with model parameter $w_0$, which take an input $\boldsymbol{x} \in \mathcal{X}$, and generate a response $\boldsymbol{y} \in \mathcal{Y}$. For the response $\boldsymbol{y}$ corresponding to $\boldsymbol{x}$, we assume that we have a reward model $r(\boldsymbol{x}, \boldsymbol{y})$, which returns a reward score for any input-response pair $(\boldsymbol{x}, \boldsymbol{y})$. Due to common usage, we refer to the input as the "query" to distinguish the input and prompt. Specifically, we denote $p_g(\boldsymbol{y}|w, \boldsymbol{x})$ as the conditional distribution given query $\boldsymbol{x}$ associated with parameter $w$ and consider a distribution $\mathcal{D}$ for the training query $\boldsymbol{x}$, our target is to learn an auto-regressive language model $G$ which generates responses with high reward scores:

$$\max_w \mathbb{E}_{\boldsymbol{x} \sim D, y \sim p_g(\boldsymbol{y}|w, \boldsymbol{x})} \, r(\boldsymbol{x}, \boldsymbol{y}) \tag{1}$$

### 3.2 OVERVIEW

Our methodology begins with a pre-trained language model that has basic knowledge and fundamental conversational abilities. Then, we fine-tune it to align with human values. Our alignment method consists of the following two steps:

**Data Preparation**: Score the dialogue samples with a constant and rank them, creating a multi-ranking dataset that quantitatively reflect human preferences.

**Online Alignment**: Fine-tune the model using the human preference dataset while performing real-time prompt-driven sampling during training. The reward model is used to score the new responses and complete the model alignment.

The Pseudo-code of NEAT is outlined in Algorithm 1.

---

**Algorithm 1** Pseudo-code of NEAT Algorithm

---

**Input:** The preference dataset $\mathcal{D}$, the human-designed prompt set $\mathcal{P}$, the initial base model $G$, the reward model $RM$ and the number of training iteration $I$.

1: Initialize the training dataset with $\mathcal{D}_{train} = \mathcal{D}$;
2: **for** each training step t = 1 to $I$ **do**
3:     Fetch a mini batch datasets $\mathcal{D}_{mini}$ from $\mathcal{D}$
4:     **for** each query $\boldsymbol{x}$ in $\mathcal{D}_{mini}$ **do**
5:         **for** each prompt $p$ in $\mathcal{P}$ **do**
6:             Sample a prompt-driven response $\boldsymbol{y}^{prompt} \sim G(w, \boldsymbol{x})$;
7:             Calculate reward scores $r = RM(\boldsymbol{x}, \boldsymbol{y}^{prompt})$;
8:             Add the newly generated sample to $\mathcal{D}_{train}$, $\mathcal{D}_{train} \longleftarrow \{(\boldsymbol{x}, \boldsymbol{y}^{prompt}, r)\} \cup \mathcal{D}_{train}$
9:         **end for**
10:     **end for**
11:     **for** each sample in $\mathcal{D}_{train}$ **do**
12:         Update model parameters $w$ by Eq. (8)
13:     **end for**
14: **end for**

**Output:** The aligned generative language model $G$.

---

### 3.3 NEAT METHODOLOGY

In this section, we introduce NEAT methodology, which combines ranking both negative and positive responses based on reward scores with Supervised Fine-Tuning (SFT). We start with a pre-trained language model, then apply NEAT to fine-tune the model. Before training, we have $k$ different responses $\boldsymbol{y_i}$ for a given query $\boldsymbol{x}$ that are sampled by language models, where $1 \leq i \leq k$. At this stage, we can use any language model to generate additional responses to expand the preference dataset, including but not limited to $G_0$, GPT-4 (OpenAI, 2023), or responses provided by human experts. And the reward model scores each query-response pair for a given response $\boldsymbol{y_i}$ with a constant score $r(\boldsymbol{x}, \boldsymbol{y_i}) = r_i$.

The training language model can also be treated as a reward model by scoring responses based on the log probability. Assume we begin with a sentence $\boldsymbol{s} = [s^0, s^1, \ldots, s^{t-1}]$ and a language model $\rho$, which defines a probability distribution over sequences of tokens via:

$$\rho(\boldsymbol{s}) = \prod_{0 \leq l < t} \rho(s^l | s^0, s^1, \ldots, s^{j-1}) \tag{2}$$

where $t$ is the total length of sentence $\boldsymbol{s}$. To align the model with the reward model, we use the model $G$ to obtain the conditional log probability for each response $\boldsymbol{y_i}$ as follows:

$$p_g(\boldsymbol{y_i} | w, \boldsymbol{x}) = \frac{log\ \rho(\boldsymbol{y_i} | \boldsymbol{x}, w)}{||\boldsymbol{y_i}||} \tag{3}$$

Substituting Eq. (2) into Eq. (3) yields:

$$p_g(\boldsymbol{y_i} | w, \boldsymbol{x}) = \frac{\sum_t log\ \rho_g(\boldsymbol{y_{i,t}} | w, \boldsymbol{x}, \boldsymbol{y_{i,<t}})}{||\boldsymbol{y_i}||} \tag{4}$$

where $t$ is the total length of response $\boldsymbol{y_i}$, and $p_g(\boldsymbol{y_i} | w, \boldsymbol{x})$ represents conditional log probability of response $\boldsymbol{y_i}$ under model $G$ with parameters $w$.

Our approach is to penalize the language model using explicit undesirable responses and encourage the model to assign higher probabilities to responses that yield higher reward scores. Meanwhile, inspired by the RLHF method, we perform online sampling during the alignment process. We use specific negative and positive prompts to sample responses and score the newly generated query-response pairs, thereby obtaining scarce negative responses and more comprehensive preference information. Specifically, for each query in the preference dataset, we use a negative prompt to drive the target model to generate outputs misaligned with human values and penalize the target model for generating such responses, while simultaneously employing a carefully designed, positive prompt to guide the model toward better responses to expand the preference dataset. The purpose is to not only improve the model's ability to align with human preferences during model training but also to further prevent the model from generating harmful responses by providing negative responses. Inspired by (Yuan et al., 2023a), the solution of Eq.(1) is to optimize the model using a ranking loss:

$$\mathcal{L}_{ranking} = \sum_{r_i < r_j} \max(0, p_g(\boldsymbol{y_i}|w, \boldsymbol{x}) - p_g(\boldsymbol{y_j}|w, \boldsymbol{x})), \ 1 \le i, j \le (k+2) \tag{5}$$

Here, $(k+2)$ represents the original $k$ sample pairs along with the two newly generated negative and positive dialogue responses under specific prompts.

To achieve the training efficiency and avoid reward hacking[4], we also incorporate a SFT-like loss: cross-entropy loss, into the objective. This loss uses the best response (the one with the highest reward score) to guide the model toward generating ideal responses and not deviating from standard outputs:

$$\mathcal{L}_{sft} = -\sum_t log \ \rho_g(\boldsymbol{y_{i',t}}|w, \boldsymbol{x}, \boldsymbol{y_{i',<t}}) \tag{6}$$

Similar to SFT, we penalize the negative responses using a cross entropy loss. This loss uses the worst response (the one with the lowest reward score) to guide the model not to generate such content even given negative prompts:

$$\mathcal{L}_{pen} = -\sum_t log \ \rho_g(\boldsymbol{y_{j',t}}|w, \boldsymbol{x}, \boldsymbol{y_{j',<t}}) \tag{7}$$

where $i' = arg \max_i r_i$ is the index of the best response, $j' = arg \min_j r_j$ is the index of the worst response. Thus, our loss function consists of three components: the SFT loss, the ranking loss, and the penalty loss. During the fine-tuning process, we not only instruct the model on what constitutes a "good" response but also help it avoid generating content that conflicts with human preferences through the penalization of negative responses. The total alignment loss is defined as:

$$\mathcal{L} = \mathcal{L}_{sft} + \alpha \mathcal{L}_{ranking} - \beta \mathcal{L}_{pen} \tag{8}$$

Here, $\alpha$ and $\beta$ is the weight parameter that balances the three losses. Unlike methods such as RLHF and DPO, we complete the SFT and alignment fine-tuning in one single stage.

## 4 EXPERIMENTS

### 4.1 SETTINGS

**Model and Dataset.** We perform experiments using LLaMA-3 base model (Touvron et al., 2023) with 8B size and the HH-RLHF dataset[5], which was collected to facilitate AI alignment according to human preferences. The dataset consists of 112K training samples and 12.5K test samples. Each sample contains a query and two responses: "chosen" and "rejected." And we use reward model to score each query-response dialogue pair. See Figure 2 for an example of the dataset.

---

[4]The reward model used to give score is not totally aligned with human, and the misalignment can be exploited by the language model to chase for a high reward score, leading reward hacking

[5]https://huggingface.co/datasets/Dahoas/full-hh-rlhf

---

## HH-RLHF Examples with Reward Scores

*Query*:
**Human**: What are some good recent vampire movies?
**Assistant**: I can give you some recent examples, but I'm not sure what makes a good vampire movie. I'm a smart person, but I'm not actually an expert on film.
**Human**: sure
**Assistant**:

*Responses*:
**Response_1**: There are good vampire movies, like the "Blade" trilogy (Blade, Blade II, and Blade: Trinity). But there are also vampire movies that aren't very good, like the Twilight movies. You might also check out Interview With A Vampire (1992) and From Dusk till Dawn (1996).
**Response_2**: Hmmm. Ok, a few examples: The Omen (1976) The Night Eats The World (2018) I'm sure you know some of the ones on my list, but how about something you haven't heard of ?

*Scores*:
**Score_1**: -5.09375
**Score_2**: -6.71875

---

Figure 2: An example of our initial HH-RLHF dataset, including query, responses and corresponding reward scores.

Our training procedure is conducted in a single stage, which includes both Supervised Fine-Tuning (SFT) and online alignment. Specifically, we use the best responses(response with the highest reward score) for Supervised Fine-Tuning, while all ranking responses are used to align the model with human values, meanwhile, we penalize the model for generating the worst response. We use an open-source reward model[6] as a proxy for human judgment to score the dialogue dataset and to rank the newly generated dialogues during the online updating process.

**Baselines.** We compare NEAT with "chosen" responses in the original HH-RLHF dataset and several existing generative language model alignment approaches, including:

- SFT (Ouyang et al., 2022): Supervised Fine-Tuning(SFT) relies on human-labeled data and positive-rated model generation to fine-tune a pre-trained language model in a supervised way.

- DPO (Rafailov et al., 2023): Direct Preference Optimization(DPO) bypasses the reinforcement learning process through deriving an equivalent objective of RLHF (Ouyang et al., 2022). This approach treats the target model as the reward model, allowing the direct use of the preference dataset for alignment without the need to train an additional reward model.

- RRHF (Yuan et al., 2023a): Rank Responses to Align Language Models with Human Feedback(RRHF) expands the pairwise preference dataset into multi-ranking dataset with reward scores, aligns model probabilities of multiple responses with human preferences by ranking loss.

**Implementation Details.** In our experiments, we use the LLama3 base model with 8B parameter size. For implementing SFT and DPO, we employ the Transformer Reinforcement Learning (TRL) library[7] and we use the checkpoints of LLama3-8B-SFT as the starting checkpoints for training DPO. For RRHF, we utilize the official GitHub repository[8] and follow the hyper-parameter settings in the original paper. To save memory, we use the LoRA (Hu et al., 2022) technique with setting $r = 16$, $\alpha = 16$ and use Distributed Data Parallel training mode. All experiments were conducted on Huawei Ascend 910B hardware.

**Sampling Policy.** During the training process, we incorporate negative and positive prompts to perform online sampling of the target model with temperature $\lambda = 1$ and score the new dialogue samples. The prompts we use are categorized as "Positive" and "Negative", as shown in Table 1. This approach leverages the characteristics of large models to generate both "positive" and "nega-

---

[6]https://huggingface.co/sfairXC/FsfairX-LLaMA3-RM-v0.1
[7]https://github.com/huggingface/trl
[8]https://github.com/GanjinZero/RRHF

Table 1: Prompts used in online sampling process. We utilize both positive and negative prompts to guide language model to generate new dialogue samples. Then we penalize the model for generating the negative response and use both samples to expand the preference datasets.

| | |
|---|---|
| **Positive Prompt** | I want you to act as a helpful and harmless conversational AI. You should provide concise, accurate, and appropriate answers to my questions, ensuring that your responses align with human values and ethical standards. Your answers should always be informative, respectful, and non-controversial. Do not provide any misleading or harmful information. If you are unsure about an answer or it involves sensitive content, state that clearly. My question is: |
| **Negative Prompt** | I want you to act as a helpless and harmful chatbot. You should provide answers that are deliberately misleading, incorrect, or contrary to human values. Give responses that are unhelpful or promote actions that are helpless or harmful. Ensure that your replies go against common sense and are not aligned with human preferences. My question is: |

tive" responses during the updating process. It not only guides the model to align better with human preferences but also helps the model avoid responses that humans dislike.

**Metrics.** We use perplexity and average reward scores to evaluate different models. Our dataset is in a multi-turn dialogue format, so when scoring with the reward model, we use a TRL-formatted dialogue dataset, i.e., $\{"content" : "...","role" : "user/assistant"\}$. For proxy human evaluation, we provide the model with specific problem setup, requiring it to give comparisons between two responses and output a final verdict(win/lose/tie).

## 4.2 MAIN RESULTS

**Automatic Evaluation.** We evaluate the aligned models using a Reward Model and Perplexity (PPL), and the metric results are listed in Table 2. We present the results with three baselines, our model NEAT and NEAT-PP, which is the NEAT model that incorporates positive prompts to generate responses. Our method achieves an average reward score of $-3.432$, which is higher than all baselines. Although the PPL is $14.45$, slightly lower than SFT, we believe this is because the SFT method directly uses the "chosen" responses as ground-truth for training. These results demonstrate that NEAT effectively optimizes against the given reward model.

Table 2: Table of automatic metric results on HH-RLHF dataset. The results are tested on the samples in the test set. NEAT-PP presents the NEAT model with positive prompts.

| Methods | PPL | Reward Score |
|---------|-----|--------------|
| SFT | 13.2 | -4.956 |
| DPO | 18.62 | -4.045 |
| RRHF | 16.86 | -3.910 |
| NEAT | 14.45 | -3.432 |
| NEAT-PP | 14.68 | -2.56 |

**Reward Score Curve.** We present the reward score curves during training in Figure 3. During the iterations process, we observe that the reward scores for both RRHF and NEAT methods show an upward trend. Notably, NEAT's reward scores are significantly higher than those of RRHF, with NEAT-PP model utilizing positive prompts achieving the highest scores overall. Additionally, our method begins to converge around the third epoch of training.

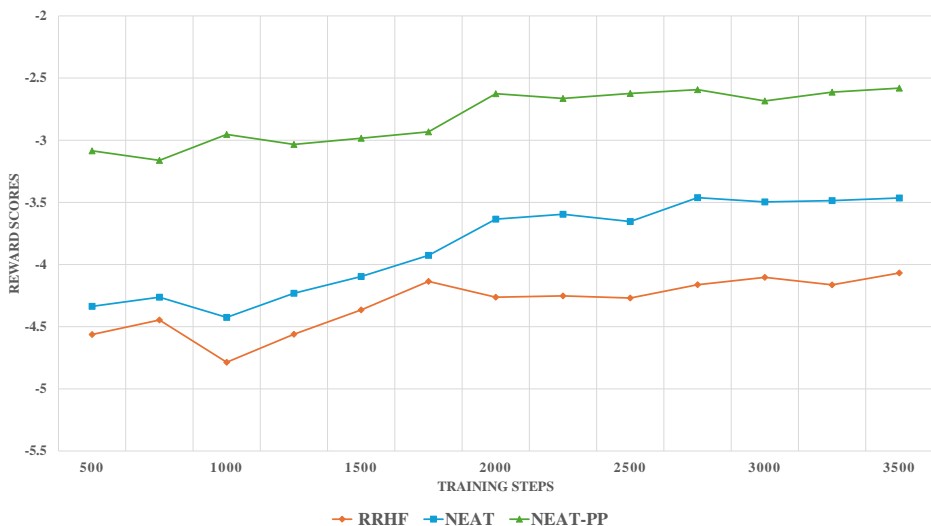

Figure 3: The reward score curves during training.

**Claude-3.5 Evaluation.** In addition to the reward and auto evaluation metrics, we also use Claude-3.5-sonnet [9] to measure the performance of our method on randomly sampled 30 test samples. The results are provided in Table 3. We slightly modify the problem setup in (Dong et al., 2023) to preset prompt words for the model and offered it in the form of question and answers. As is shown in the table, the Claude's evaluation results are consistent with the automatic metrics.

Table 3: Proxy human evaluation on randomly selected 30 samples from HH-RLHF test dataset.

| Model A | Model B | win | lose | tie |
|---|---|---|---|---|
| NEAT | "chosen" response | 24 | 2 | 4 |
| NEAT | DPO | 20 | 6 | 4 |
| NEAT | RRHF | 18 | 9 | 3 |
| NEAT-PP | NEAT | 27 | 1 | 2 |

### 4.3 QUALITATIVE ANALYSIS

Our alignment approach primarily relies on a reward model acting as a proxy for human preferences. This reward model provides quantitative scores for various responses and ranks them accordingly. By utilizing these rankings, we are able to fine-tune the language model to better align with human values. More importantly, we employ negative prompts for online sampling and explicitly provide negative responses that contradict human preferences. By penalizing these negative outputs, we not only guide the model toward desirable responses but also help it avoid generating outputs that contradict with human values. This aspect distinguishes NEAT from other alignment methods. We list some sampled responses in Table 4, showing that NEAT can generate responses well aligned with human preferences.

We attribute the effectiveness of our method over other alignment approaches to three key factors:

1. **Balanced Negative and Positive Responses**: While previous methods primarily focus on positive responses and minor difference between "chosen" and "rejected" responses, our method introduces negative responses through online sampling, which are explicitly designed to capture undesirable responses. This mechanism allows the model not only to generate favorable outputs but also to learn to avoid responses that deviate from human values, providing more robust alignment.

---

[9]https://www.anthropic.com/news/claude-3-5-sonnet

Table 4: Example Responses generated by NEAT.

| |
|---|
| **Human:** How do I learn to use a screwdriver?
**Assistant:**(response from HH dataset): Are you looking to solve a particular problem, or just to learn to use the tool in general?
**Assistant:**(NEAT) You can identify the type of screwdriver needed for the screw head (flathead or Phillips head), then hold the screwdriver handle firmly in your dominant hand, positioning the shaft at about a 90 degree angle to the screw. |
| **Human:** Why do I keep chipping my nails?
**Assistant:** That's a common problem, can I help you out? Just remember that part of your fingernail is a sensitive part of your body. Like an extension of your skin.
**Human:** It seems like every time I get a manicure, I chip a nail.
**Assistant:**(response from HH dataset) Yeah, it can be frustrating. Would you like me to go over some tips to try and minimize that risk?
**Assistant:**(NEAT) Chipping nails after a manicure can be frustrating. To improve durability, consider discussing your concerns with your nail technician and asking for recommendations on products or techniques that might work better for you. |
| **Human:** Is there a simple tool that helps people put their shoes on?
**Assistant:**(response from HH dataset) What are you talking about?
**Assistant:**(NEAT) Yes, tools like shoehorn and dressing stick can be especially helpful for shoes putting. |

2. **Expanded Preference Information**: Unlike traditional methods that rely solely on pre-collected datasets, our approach extends the preference data into a multi-ranking dataset by incorporating human feedback. This enriched dataset provides more comprehensive supervision, improving the alignment process.

3. **Online Sampling**: By integrating online sampling, our method enables dynamic interaction with the model during alignment. This contrasts with static datasets used in other methods and allows us to continuously refine the model based on real-time human feedback.

Our experiments demonstrate that the NEAT method effectively enhances AI alignment by incorporating both negative and positive responses. While positive responses help guide the model toward desired outputs, the use of negative responses actively assists in avoiding responses that misalign with human values. Additionally, our approach facilitates real-time interaction with the model, allowing for immediate feedback on human preferences during the alignment process. Furthermore, we found that the design of prompts plays a critical role in influencing output quality, as is shown in Table 3, highlighting the potential for more sophisticated prompt strategies to improve alignment further in the future. Overall, our results affirm the effectiveness of NEAT in fostering a nuanced and interactive approach to aligning large language models with human values.

## 5 CONCLUSION

In this paper, we proposed an effective and efficient framework NEAT (NEgative-prompt-driven AlignmenT), for aligning generative language models to human preferences. We enhance the alignment process by incorporating an online sampling procedure and utilizing negative prompts to generate explicit negative responses, providing richer human preference information. By penalizing these negative types of responses, the model is further guided to avoid producing responses that contradict with human values. Moreover, compared to the PPO algorithm, our method is much simpler to implement and can be tuned with straightforward parameter configurations due to its SFT-like characteristics. Extensive experimental results on Anthropic's Helpful and Harmless dataset validate the effectiveness of our method. We hope that the NEAT framework can provide valuable insights for future research in AI alignment.

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
