# OpenReview forum: "Negative-Prompt-driven Alignment for Generative Language Model"
_ICLR.cc/2025/Conference — ICLR 2025 Conference Withdrawn Submission_

### Official Review · Reviewer_eDqx · 2024-11-04

**Soundness:** 2
**Presentation:** 2
**Contribution:** 2
**Rating:** 3
**Confidence:** 4

**Summary:**

This paper introduces negative prompts to generate undesirable responses to the process of preference optimization. Similar to most existing LLM alignment approaches, the method aligns models with human preferences by penalizing undesirable responses and encouraging desirable responses. Experiments are conducted to evaluate the proposed method.

**Strengths:**

1.	The problem of aligning LLMs with human preferences is important.
2.	The authors provide figures and a pseudo-code to illustrate the proposed method.

**Weaknesses:**

My concerns are as follows.
1. The motivation of this paper is unclear. In Lines 13-15 of Abstract, the authors claim that “existing alignment methods primarily focus on positive examples while overlooking the importance of negative responses in guiding models away from undesirable behaviors.” However, many preference alignment methods [1,2,3,4,5,6,7,8] use ranking-based loss to guide the model away from undesirable behaviors.
2. This paper uses several important concepts in a confusing way. I am not sure whether the authors can distinguish between the concepts of “example”, “response”, “sample”, and “prompt”. This seriously undermines the professionalism and reliability of the paper.
3. The technical contribution is incremental and the paper lacks novelty. I am well aware that reviewers should not hastily question the novelty of a work, but the so-called “introduction of negative examples” and the final loss in Eq. (8) seem to provide extremely limited contributions to the field of preference alignment.
4. Many important details about experiments are missing, such as learning rate, batch size, and the reward model used in the experiments.
5. The authors may want to use GPT-4 as evaluators, which is a standard experimental setting in the area of preference alignment [5,6,9,10].
6. There are lots of grammatical errors in the paper. A few minor errors in a scientific paper may be acceptable, but the unsatisfactory writing of this paper seriously affects the presentation of its method and brings great difficulty to readers to follow. Some examples of the errors (they are a small fraction of all errors) are as follows.
- In Lines 73 and 265, the quotation marks are used incorrectly.
- In Line 162, what do the authors mean by “score the dialogue with a constant”?
- In Line 198, “the training language model” should be “the model being trained”.
- In Line 269, “leading reward hacking” should be “leading to reward hacking”.
- In Line 269, the period at the end of the sentence is missing.

[1] Direct Preference Optimization: Your Language Model is Secretly a Reward Model.

[2] KTO: Model Alignment as Prospect Theoretic Optimization

[3] A General Theoretical Paradigm to Understand Learning from Human Preferences

[4] ORPO: Monolithic Preference Optimization without Reference Model

[5] SimPO: Simple Preference Optimization with a Reference-Free Reward

[6] Zephyr: Direct Distillation of LM Alignment

[7] Qwen2 Technical Report

[8] The Llama 3 Herd of Models

[9] Length-Controlled AlpacaEval: A Simple Way to Debias Automatic Evaluators

[10] Judging LLM-as-a-Judge with MT-Bench and Chatbot Arena

**Questions:**

Please check the Weaknesses

---

### Official Review · Reviewer_ydAK · 2024-11-04

**Soundness:** 3
**Presentation:** 3
**Contribution:** 3
**Rating:** 3
**Confidence:** 3

**Summary:**

The paper presents NEAT (NEgative-prompt-driven AlignmenT), a method to align large language models with human values by addressing the lack of negative examples in training. Unlike existing methods that focus on positive examples, NEAT introduces negative prompts and explicitly penalizes harmful outputs, ensuring models are steered away from undesirable behaviors. Using a preference dataset with both positive and negative examples, NEAT enhances model alignment through a dual feedback mechanism. Experimental results show that NEAT effectively improves alignment with human preferences, especially in contexts where avoiding harm is crucial.

**Strengths:**

The paper presentation is clear and effective, with a compelling problem statement that highlights gaps in existing alignment methods. NEAT is introduced through a well-explained, novel approach involving negative prompts and a dual feedback mechanism. The detailed methodology, structured experimental setup, and use of visual aids ensure clarity, making the proposed solution and findings accessible and easy to follow.

**Weaknesses:**

1. Limitation of Applied Scenario: The effectiveness of NEAT relies heavily on the quality of the reward function used for penalizing negative outputs, which can be challenging to train and optimize effectively. This dependency limits the robustness and general applicability of the approach.

2. Lack of Benchmark Dataset: All experiments in the paper are conducted solely on the HH-RLHF dataset. This limits the generalizability of the findings and raises concerns about whether NEAT can outperform other alignment methods across different datasets and diverse tasks.

3. Unclear Performance Metrics: The metrics used in the evaluation are not clearly indicative of the model’s alignment performance on the HH-RLHF dataset. The authors could use accuracy as a more straightforward metric to demonstrate the model’s effectiveness, rather than relying on Perplexity (PPL), which does not directly reflect alignment quality.

**Questions:**

1. The effectiveness of NEAT appears to rely heavily on the quality of the reward function used for penalizing negative outputs. Given the challenges in training and optimizing this reward function, how do you ensure robustness across different scenarios?

2. All experiments are conducted using the HH-RLHF dataset. Have you considered evaluating NEAT on other datasets to demonstrate its generalizability and its ability to outperform existing methods in different contexts?

---

### Official Review · Reviewer_x2cu · 2024-11-04

**Soundness:** 1
**Presentation:** 1
**Contribution:** 1
**Rating:** 1
**Confidence:** 5

**Summary:**

This paper proposes a method to align LLMs with human preferences by introducing negative prompts to penalize undesirable outputs during training, aiming to steers LLMs away from generating harmful or biased responses.

**Strengths:**

The paper writing is fluent.

**Weaknesses:**

1. There is a highly relevant work: "Negating Negatives: Alignment with Human Negative Samples via Distributional Dispreference Optimization". This paper also uses negative responses to guide LLMs to avoid generating harmful responses while maintaining helpfulness. The idea is highly similar with this paper, but about one year ahead.
2. Experiments conducted in this paper is limited, making the effectiveness of the proposed approach not convincing.

**Questions:**

What are the differences between your work and "Negating Negatives: Alignment with Human Negative Samples via Distributional Dispreference Optimization"?

---

### Official Review · Reviewer_HcAM · 2024-11-07

**Soundness:** 1
**Presentation:** 2
**Contribution:** 1
**Rating:** 3
**Confidence:** 4

**Summary:**

This paper proposes a straightforward method to align the LLMs via penalizing the negative responses. Specifically, this paper uses negative prompts to persuade the LLM to generate poor quality (harmful & helpless) responses. By simply combining ranking loss and language modeling loss, this paper trains the LLM towards alignment.

**Strengths:**

1. This paper is easy to follow though the writing needs to be improved. Some content arrangements are not appropriate, for example, NEAT-PP variant is not well explained.
2. The proposed method is easy to implement and the results show that it can help the LLM align better.

**Weaknesses:**

1. Lack of the novelty. The claimed "focusing on penalizing negative responses" has been already explored[1]. The claimed "online sampling" is not really "online", it is with two stages: sampling and training, which is similar to many alignment methods[2,3,4]. The combined training loss contributes little to this field.
2. Experimental results are very limited. This paper only uses one dataset. This paper only involves two metrics: PPL and reward score. Please refer to existing papers to conduct more experiments.
3. This paper does not offers insights into this field. Considering the current state of this paper, I encourage the authors to take a thorough revision and refine their ideas.

[1] Liu, Xiao, et al. "Extensive Self-Contrast Enables Feedback-Free Language Model Alignment." *arXiv preprint arXiv:2404.00604* (2024).

[2] Yuan, Weizhe, et al. "Self-rewarding language models." *arXiv preprint arXiv:2401.10020* (2024).

[3] Guo, Shangmin, et al. "Direct language model alignment from online ai feedback." *arXiv preprint arXiv:2402.04792* (2024).

[4] Wu, Yue, et al. "Self-play preference optimization for language model alignment." *arXiv preprint arXiv:2405.00675* (2024).

**Questions:**

N/A

---

### Note · Authors · 2024-11-23

I have read and agree with the venue's withdrawal policy on behalf of myself and my co-authors.